# MP-Nav: Enhancing Data Poisoning Attacks against Multimodal Learning

**Jingfeng Zhang** [1]  **Prashanth Krishnamurthy** [1]  **Naman Patel** [1]  **Anthony Tzes** [1]  **Farshad Khorrami** [1]

## Abstract

Despite the success of current multimodal learning at scale, its susceptibility to data poisoning attacks poses security concerns in critical applications. Attacker can manipulate model behavior by injecting maliciously crafted yet minute instances into the training set, stealthily mismatching distinct concepts. Recent studies have manifested the vulnerability by poisoning multimodal tasks such as Text-Image Retrieval (TIR) and Visual Question Answering (VQA). However, the current attacking method only rely on random choice of concepts for misassociation and random instance selections for injecting the poisoning noise, which often achieves the suboptimal effect and even risks failure due to the dilution of poisons by the large number of benign instances. This study introduces Multimodal Poison Navigator (MP-Nav), a plug-and-play module designed to evaluate and enhance data poisoning attacks against multimodal models. MP-Nav operates at both the concept and instance levels, identifying semantically similar concept pairs and selecting robust instances to maximize the attack efficacy. The experiments corroborate MP-Nav can significantly improve the efficacy of state-of-the-art data poisoning attacks such as AtoB and ShadowCast in multimodal tasks, and maintain model utility across diverse datasets. Notably, this study underscores the vulnerabilities of multimodal models and calls for the counterpart defenses.

## 1. Introduction

Multimodal learning (Yuhas et al., 1989; Hall & Llinas, 1997; Ngiam et al., 2011; Andrew et al., 2013; Baltrusaitis et al., 2019) integrates data from multiple modalities (such as vision, text, audio, and sensory signals) and outputs mul-

[1]New York University. Correspondence to: Farshad Khorrami <khorrami@nyu.edu>.

*Proceedings of the 42nd International Conference on Machine Learning*, Vancouver, Canada. PMLR 267, 2025. Copyright 2025 by the author(s).

timodal models that have recently achieved excellent performance in many tasks (Radford et al., 2021). Specifically, current multimodal models have greatly advanced many tasks including image captaining, visual question answering, text-image retrieval with some salient examples such as CLIP (Radford et al., 2021), ALBEF (Li et al., 2021), LLaVA (Liu et al., 2023), and commercialized models (OpenAI, 2024; Gemini Team, 2024).

Despite the extraordinary performance, recent studies (Carlini & Terzis, 2022) have exposed security vulnerabilities in current multimodal learning, urging the need for meticulous caution in its critical applications to robotics (Bhat et al., 2024a;b; Wu et al., 2024), medicine (Jin et al., 2024), etc. To comprehensively uncover the risks, researchers have holistically evaluated vision-language multimodal models (Lee et al., 2023; 2024), and developed attacks against the multimodal models, such as evasion adversarial attacks (Lu et al., 2023; He et al., 2023; Wang et al., 2023; Bai et al., 2024; Gao et al., 2025) and jailbreaking attacks (Qi et al., 2024; Shayegani et al., 2024; Tao et al., 2024). In evasion attacks, the *adversarial users* can maliciously manipulate the inputs (but without affecting model parameters), and cause multimodal models to produce incorrect predictions. In contrast, another less studied direction is data poisoning attack (Biggio et al., 2012), where model parameters can be tempered by the injection of a small amount of poisoned data into the training set, and the manipulated models affect the vast majority of *benign users* who query the model normally. Even worse, modern multimodal learning often relies on a large amount of noisy and uncrated data from external sources, which exacerbates the threat of data poisoning attack.

There are three pioneering studies on data poison attacks against multimodal learning (Carlini & Terzis, 2022; Yang et al., 2023; Xu et al., 2024). Carlini & Terzis (2022) has shown that multimodal models are more vulnerable to data poisoning attacks than unimodal models, and even worse, the vulnerability increases with model capacity because larger multimodal models can memorize more incorrect associations between different concepts. Furthermore, Yang et al. (2023) extended the data poisoning attacks to the CLIP-powered test-image retrieval (TIR) tasks (Radford et al., 2021), where given texts from original concept $\mathcal{O}$, the poisoned CLIP will mistakenly and consistently retrieve the images from targeted concept $\mathcal{T}$. Besides, Xu

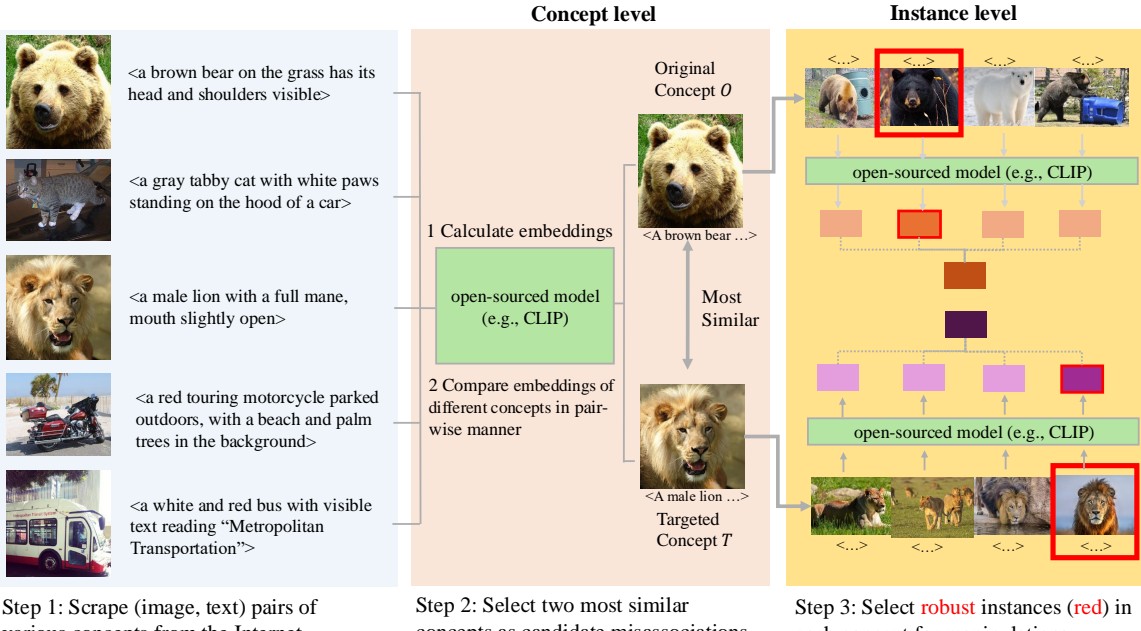

*Figure 1.* The MP-Navigator (MP-Nav) is a plug-and-play module to evaluate (or even enhance) the existing poisoning attack methods. The MP-Nav selects similar concepts for the candidate misassociations and then selects robust instances to inject the poisoning noises.

et al. (2024) showed the data poison attacks can maliciously affect Vision-Question Answering (VQA) tasks (Gurari et al., 2018; Hudson & Manning, 2019), where the poisoned vision-language model (VLM) will mistakenly recognize images from concept $\mathcal{O}$ and generate misleading descriptions of concept $\mathcal{T}$, As the attacker desires, the poisoned multimodal model will stealthily associate two distinct concepts (original concept $\mathcal{O} \rightarrow$ targeted concept $\mathcal{T}$) without degrading overall utility, thus threatening the large number of the benign users.

Although the threat has been demonstrated, a gap remains in enhancing the data poisoning attacks. First, not all concepts are equally vulnerable to the misassociation. For example, an attacker attempting to misassociate the concept of "Bear" with "Bus" would face a greater challenge compared to a misassociation between "Bear" and "Lion". "Bear" and "Lion" often share greater semantic and visual similarities, making this misassociation more feasible. Second, not all instances contribute equally. Xia et al. (2022); Gao et al. (2023) studied backdoor attacks against unimodal models and unveiled that different instances play unequal roles in contributing to the attack efficacy. Randomly selecting instances to inject poisoning noise often leads to suboptimal results, requiring a significantly larger amount of poisoned instances to achieve the desired effect; otherwise, the poisoning attack is likely to fail due to the dilution of the poisoning noise by the benign instances. For example, (Xu et al., 2024) showed a misaacoication between two distinct con-

cept "Biden" and "Trump" in a VQA task, but achieving this required poisoning all instances of the concept "Biden" in the training data to ensure consistent manipulation. This exhaustive approach negates the fact benign instances can significantly dilute the poisoning noise and make the attacks less effective.

Therefore, given practical scenarios where attackers can only manipulate a limited number of instances and have no control over the number of benign instances mixed into the training data, attackers would significantly benefit from a systematic guidance framework to select source/target concepts and corresponding instances to construct effective data poisoning attacks. To this end, this paper proposes a plug-and-play module Multimodal Poison Navigator (**MP-Nav**) that guides the attacker to craft effective multimodal poisoning attacks at the concept and instance levels. As shown in Figure 1, MP-Nav facilitates the attack by first identifying the most similar concepts that can ease the attacker's effort to misassociate. MP-Nav achieves this by scraping (image, text) instances of various concepts from publicly available sources and comparing their semantic similarities within the model's embedding space. By leveraging the embedding space, MP-Navigator ensures that selected target and source concepts share high latent feature similarity. Furthermore, instance-level MP-Nav ranks and identifies the most robust instances within the chosen concepts. By poisoning the robust instances, the attacker can largely resist the delusion of the large majority of benign instances. Experimental

results on real-world TIR and VQA tasks corroborates our proposed MP-Nav that can indeed significantly improve the efficacy of data poisoning attacks against multimodal models, while preserving the model utility.

## 2. Preliminary

### 2.1. Multimodal Learning

A training set $S = \{(x_1, y_1), ..., (x_n, y_n)\}$ is a finite set of instance pairs of two modalities, where $x$ refers to images, and $y$ refers to text captions throughout this paper. The learner takes input the training set $S$ and outputs encoders $\varepsilon$ of different modalities. To customize encoders on different tasks, and a fusion model $f$ is fine-tuned on a small dataset.

In the multimodal contrastive learning (Li et al., 2021; Radford et al., 2021; Li et al., 2023), the learner jointly trains an image encoder $\varepsilon_{img}$ and text caption encoder $\varepsilon_{txt}$ that encode the original images or texts into low-dimensional embeddings. The learning objective is to maximize the embeddings similarity between $< \varepsilon_{img}(x_i), \varepsilon_{txt}(y_i) >$ of the same instance $i$ and at the same time minimize the similarity between embeddings $< \varepsilon_{img}(x_i), \varepsilon_{txt}(y_j) >$ of different instances ($i \neq j$). The learner typically pretrains encoders on massive datasets covering diverse domains, and thus the learned encoders have some zero-shot generalization ability (Brown et al., 2020; Min et al., 2022). Then, the well pre-trained encoders are typically used in two ways.

In the zero-shot way, the parameters of multimodal encoders are frozen and not explicitly trained on task-specific examples. One typical example is the Text-Image Retrieval (TIR) task (Cao et al., 2022). Given a text description (e.g., $y = $ "a photo of bear"), TIR will evaluate embeddings of all images $\varepsilon_{img}(\cdot)$ from the database, compare those with $\varepsilon_{txt}(y)$, and retrieve several "bear" images whose embeddings are closest to $\varepsilon_{txt}(y)$. TIR task is commonly used in the current image search engine.

In the fine-tuning way, the parameters will be finetuned using task-specific instances, and a fusion model is optimized on the top of encoders to adapt to various tasks. For example, in the Visual Question Answering (VQA) task, the pre-trained image and text encoders are leveraged to extract embeddings, and a fusion model $f(\varepsilon_{img}(\cdot), \varepsilon_{txt}(\cdot))$ is trained based on a small and customized VQA dataset. If the VQA task is predicting from a fixed set of answers, we could use cross-entropy loss $\ell_{CE} = - \sum_i z_i log(f(\cdot)_i)$, where $z_i$ is the ground-truth answer and $f(\cdot)$ returns the predicted answer. For the open-ended VQA, we can train the fusion model as a sequence generator using cross-entropy loss over words.

### 2.2. Poisoning Attacks against Multimodal Learning

In the poisoning attack (Biggio et al., 2012), an attacker can manipulate the training set $S$ by injecting a small amount of manipulated set $S' = \{(x'_1, y'_1), ..., (x'_m, y'_m)\}$ to form a poisoned dataset $S_p = S \cup S'$, where $m \ll n$. The data poisoning attack acts like a rumor: a small amount of poisoned instances incurs a cascading effect that can disproportionately affect various multimodal models. For example, Carlini & Terzis (2022) have showned by poisoning just three samples out of the 3 million examples in the conceptual captions dataset (Sharma et al., 2018), there is around $40\%$ probability of fooling 32 multimodal models to misassociate two distinct concepts.

**Attacker's Objectives.** Through injecting the manipulated instances, the attacker aims to misassociate two distinct concepts (i.e., $\mathcal{O} \rightarrow \mathcal{T}$) in the multimodal models without affecting the overall utility. For different tasks, the attacker could have slightly different objectives. In the TIR task, Yang et al. (2023) proposed the "AtoB" attack that given the text of "Bear" (of original concept $\mathcal{O}$), the poisoned model will retrieve a list of images that mistakenly include the "Lion" image (of targeted concept $\mathcal{T}$). In the VQA task, Xu et al. (2024) proposed the Shadowcast that can manipulate the multimodal models to misinterpret the images of original concepts to the targeted concepts $\mathcal{T}$. For example, given a photo of "Bear" (of original concept $\mathcal{O}$), the poisoned model will consistently and mistakenly output text captions related to "Lion" (of targeted concept $\mathcal{T}$).

**Attacker's Capabilities.** Following the assumptions by Biggio et al. (2012), the attacker can inject a small amount of manipulated instances into a training set. In the most practical black-box setting, the attacker has no control over the model's training (i.e., model structures, training epochs, etc) and the inference phase (i.e., model queries by the benign users).The attacker has access to all open-sourced resources (e.g., open-sourced datasets scraped from the Internet) and open-sourced models from Hugging Face). Kindly note that the setting of data poisoning attack is different from that of the backdoor attack (Gu et al., 2017; Jia et al., 2022; Bai et al., 2024), where the attacker can trigger the training data, control the model training and invoke the trigger at test time. To avoid confusions, we provided the following Table 1 for a quick comparison.

| Attacker Capacity | Poison Training Data? | Control Learning? | Trigger Test Data? |
|---|---|---|---|
| Data Poison | ✓ | ✗ | ✗ |
| Backdoor | ✓ | ✓ | ✓ |

*Table 1.* The data poisoning and backdoor attacks

For different tasks in multmodel learning, researchers have proposed different methods of poisoning attacks. Here, we review two typical methods—AtoB (Yang et al., 2023) for TIR task and Showdowcast (Xu et al., 2024) for VQA task.

### 2.2.1. AᴛᴏB Aᴛᴛᴀᴄᴋ

The AtoB method is a targeted poisoning attack for TIR tasks. The attacker aims to misassociate two distinct concepts: the original concept $\mathcal{O}$ (e.g., "bear") and the targeted concept $\mathcal{T}$ (e.g., "lion").

To achieve this, the poisoned dataset includes pairs where texts from $\mathcal{O}$ are intentionally paired with images from $\mathcal{T}$. To be specific, the attacker gathers from the clean dataset the texts describing the original concept $\mathcal{O}$ (e.g., "a brown bear standing in the grass") and also images representing the targeted concept $\mathcal{T}$ (e.g., a lion resting under a tree). Then, the attackers constructs new pairs by taking texts from $\mathcal{O}$ and intentionally pairing them with images from $\mathcal{T}$, e.g., (Text: "a brown bear standing in the grass", Image: Lion). The poisoned pairs are injected into the clean training dataset, typically in small proportions. When the multimodal model is trained on the poisoned dataset, it learns an incorrect association between $\epsilon_{\text{txt}}(y)$ (text embedding of $\mathcal{O}$) and $\epsilon_{\text{img}}(x)$ (image embedding of $T$). Consequently, when a text description related to $\mathcal{O}$ is used as a query, the poisoned model mistakenly retrieves images associated with $\mathcal{T}$. This method enforces the embeddings' misalignment in the learned space to achieve the misassociation, while preserving the correct alignments for other concept pairs.

### 2.2.2. Sʜᴀᴅᴏᴡᴄᴀꜱᴛ Aᴛᴛᴀᴄᴋ

The Shadowcast method is a clean-label poisoning attack for VQA tasks. The attacker's goal is to manipulate the model such that images associated with the original concept $\mathcal{O}$ are consistently misinterpreted as belonging to the targeted concept $\mathcal{T}$. This is achieved by injecting poison samples where the image embeddings $\epsilon_{\text{img}}(x)$ of $\mathcal{T}$ are perturbed to align with the latent features of $\mathcal{O}$, while pairing these images with text captions $y$ explicitly describing $\mathcal{T}$. The poisoned samples appear visually and textually congruent, making them undetectable during manual inspection.

Specifically, for a given image $x_{\mathcal{O}}$ representing the original concept $\mathcal{O}$ and a target image $x_{\mathcal{T}}$ representing the targeted concept $\mathcal{T}$. The attacker perturbs $x_{\mathcal{T}}$ to create a new poison image $x_p$ such that

(a) $x_p$ is visually indistinguishable from $x_{\mathcal{T}}$ (i.e., maintaining the clean-label property).
(b) The embedding $\epsilon_{\text{img}}(x_p)$ in the latent feature space is close to $\epsilon_{\text{img}}(x_{\mathcal{O}})$, aligning it with targeted concept $\mathcal{O}$.

The optimization is mathematically denoted as

$$\min_{x_p} \|\epsilon_{\text{img}}(x_p) - \epsilon_{\text{img}}(x_{\mathcal{O}})\|_2, \quad \text{subject to } \|x_p - x_{\mathcal{T}}\|_\infty \leq \rho, \tag{1}$$

where $\rho$ is the perturbation budget controlling the maximum allowable difference between $x_p$ and $x_{\mathcal{T}}$ in pixel space,

$\| \cdot \|_2$ measures the distance between embeddings in the latent space, and $\| \cdot \|_\infty$ ensures the perturbation is imperceptible to human observers. Xu et al. (2024) used projected gradient descent (PGD) method (Madry et al., 2018) and open-sourced image embedding $\epsilon_{\text{img}}$ for solving the above constrained optimization problem.

After generating $x_p$, the attacker pairs it with a text caption $y_{\mathcal{T}}$ explicitly describing the targeted concept $\mathcal{T}$. For example, if $\mathcal{T}$ is "lion", the caption $y_{\mathcal{T}}$ might be "A majestic lion resting in the grass". The generated poisoned samples are $(x_p, y_{\mathcal{T}})$, where text $y_{\mathcal{T}}$ reinforces $\mathcal{T}$, poisoning the model to associate the poisoned image with the targeted concept.

## 3. Methodology

In this section, we propose the plug-and-play MP-Nav module. MP-Nav has two important components—(1) Concept-level Selection for identifying candidate concept pairs for misassociation and (2) Instance-level Selection for selecting a few robust instances for manipulation. The two components can complement each other, which can reliably evaluate (or even enhance) the existing data poisoning attacks.

---

**Algorithm 1** Meta Algorithm of MP-Nav.

---

**Input:** Open-sourced dataset $D = \{(x_i, y_i)\}_{i=1}^N$ where $x_i$ denotes images and $y_i$ denotes text captions, open-sourced multimodal encoders $\epsilon_{\text{img}}(\cdot)$ and $\epsilon_{\text{txt}}(\cdot)$, and the attacker budget $\eta$.
**Output:** Dataset $D_p$ containing the poisoned instances.
**Component 1: Concept-level Selection**
**Step 1:** Compute mean embeddings for each concept $C_k$ in $D$ and construct a similarity matrix $S$.
**Step 2:** Identify concept pairs $(C_{\mathcal{O}}, C_{\mathcal{T}})$ of high similarity.
**Component 2: Instance-level Selection**
**Step 1:** Compute instance proximity to the concept center.
**Step 2:** Select Top-$\eta$ robust instances for manipulation.

**Execute Poisoning Manipulations**
**AtoB Attack:** Pair $x_{\mathcal{O}}$ with text captions $y_{\mathcal{T}}$ of $C_{\mathcal{T}}$, and replace $(x_{\mathcal{O}}, y_{\mathcal{O}})$ with $(x_{\mathcal{O}}, y_{\mathcal{T}})$ to form a poisoned set $D_p$.
**Shadowcast Attack:** Perturb $x_{\mathcal{T}}$ to create $x_p$ according to Eq. (1) and then pair $x_p$ with refined caption of $y_{\mathcal{T}}$, and replace $(x_{\mathcal{T}}, y_{\mathcal{T}})$ with $(x_p, y_{\mathcal{T}})$ to form a poisoned set $D_p$.

---

**Concept-Level Selection**   MP-Nav firstly identifies suitable concept pairs for misassociation. The attacker calculates the mean embeddings of both image and text of each concept of the open-sourced dataset $D$:

$$e_k^{\text{img}} = \frac{1}{|C_k|} \sum_{(x_i, y_i) \in C_k} \epsilon_{\text{img}}(\cdot), \ e_k^{\text{txt}} = \frac{1}{|C_k|} \sum_{(x_i, y_i) \in C_k} \epsilon_{\text{txt}}(\cdot),$$

where "$\cdot$" refers to either image $x_i$ or text captain $y_i$.

Next, a similarity matrix $S$ is constructed to quantify the semantic similarities between all concept pairs:

$$S[k, l] = \text{cosine\_similarity}(e_k^{\text{img}}, e_l^{\text{txt}}).$$

Instructed by this matrix, attacker can select candidate pairs of original and target concepts such that their similarity score $S[C_{\mathcal{O}}, C_{\mathcal{T}}]$ is relatively highest.

**Instance-Level Selection** We leverage a center-based selection mechanism and identifies "robust samples" that are most representative of their respective concepts in the embedding space. Instance-Level selection ensures that poisoned instances can effectively disrupt the association between original and targeted concepts, maximizing the impact of the data poisoning attack while minimizing the injection ratio.

First, we compute instance proximity to the concept center. 1. For each concept $C_k \in \{C_{\mathcal{O}}, C_{\mathcal{T}}\}$, compute the concept center embedding.

$$\mathbf{c}(C_k) = \frac{1}{|C_k|} \sum_{i \in C_k} \epsilon_i,$$

where $\epsilon_i = \lambda \cdot \epsilon_{\text{img}}(x_i) + (1 - \lambda) \cdot \epsilon_{\text{txt}}(y_i).$

2. For each instance $i$ of cencept $C_k$, compute the proximity of the instance to the concept center:

$$\text{Proximity}(i, C_k) = \text{cosine\_similarity}(\mathbf{c}(C_k), \epsilon_i).$$

3. Rank all instances in $C_k$ based on proximity scores in descending order.

Second, we select top-$\eta$ robust instances. In particular, from the ranked instances of $C_O$ and $C_T$, select the top-$\eta \cdot |C_O|$ instances closest to their respective concept centers as robust instances for manipulations.

MP-Nav can effectively evaluate (or sometimes enhance) the efficacy of existing poisoning attacks, such as AtoB and ShadowCast attacks. For the AtoB attack, the reasonable selection of robust samples ensures that the poisoned dataset enforces stronger misassociations between text and images, improving retrieval confusion in TIR tasks. For the ShadowCast attack, the focus on robust instances ensures that the perturbed samples align more effectively with the desired latent features, enhancing the attack's ability to misclassify images consistently in VQA tasks.

## 4. Experiment

The MP-Nav is a plug-and-play module that can guide the poisoning attacker to effectively select concepts and instances. In this section, we experimentally use MP-Nav to evaluate (or enhance) the existing attacks—AtoB (Yang et al., 2023) and ShadowCast (Xu et al., 2024) attacks.

### 4.1. MP-Nav for AtoB Attack against TIR Task

**Dataset** For the TIR task, we followed the previous the study of AtoB attack (Yang et al., 2023) and chose the COCO dataset (Lin et al., 2014) and Flickr-PASCAL dataset (Young et al., 2014; Rashtchian et al., 2010). COCO dataset has 80 object categories and contains 5 captions per image. For each image, we randomly selected one of the object categories as its label (a.k.a. concept). To make the COCO training set balanced, two concepts "toaster" (with 28 images) and "hair drier" (with 53 images) are removed. We used 119387 images with their corresponding captions for training and the rest 3900 images for evaluation of both model utility and poisoning efficacy.

Flickr-PASCAL is a combined dataset. Flickr dataset (Young et al., 2014) has no ground-truth concept labels but a large number of image-text pairs. In contrast, the PASCAL dataset (Rashtchian et al., 2010) is a small and balanced dataset but has ground-truth concept labels. PASCAL dataset has 1000 images with 20 labels with each image paired with 5 text captions. We divide the PASCAL dataset by half with 500 images used for injection of poisoning noises and 500 images used for evaluation of poisoning efficacy. Thus, we had 500 (from PASCAL dataset) plus 29000 images (from the Flickr dataset) used for training and 1000 images (from the Flickr dataset) for evaluation of model utility.

**Evaluation Metrics** The attacker's goal is to misassociate two distinct concepts without compromising the model's utility. To evaluate this, we use two metrics: *model utility* and *poisoning efficacy*.

To evaluate model utility, we followed the common practice (Li et al., 2021; 2022; 2023) and report R@1, R@2 and R@10 (Recall at 1, 2 and 10) scores of text-retrieval (**TR**) and image-retrieval (**IR**). R@K measures the proportion of queries for which the ground-truth item is ranked within the top K retrieved results. Specifically, the COCO test set contains 3,900 images with captions, and the Flickr-PASCAL test set has 1000 images with captions. For TR score, given a test image, we rank all caption descriptions based on their cosine similarities to the image in the embedding space, and check if the ground-truth captions appear within the top K results, and R@K reports the fraction of the test images whose ground-truth captions are in the top K results. Likewise, for IR score, we give a caption and compare all candidate images. Note that ground-truth labels (concepts) are not needed for utility evaluation.

To evaluate poisoning efficacy, we need ground-truth labels with the COCO test set of 3,900 images but a new Flickr-PASCAL test set of 500 images (a subset of the PASCAL dataset). The attacker's goal is misassociate the original concept with the targeted concept ($\mathcal{O} \rightarrow \mathcal{T}$). Following

| Tasks | Similarity | Hit@1 ↑ | Hit@5 ↑ | Hit@10 ↑ | Min Rank ↓ |
|---|---|---|---|---|---|
| boat2dog(Baseline) | 0.238 | 0.010(±0.006) | 0.214(±0.039) | 0.505(±0.069) | 15.705(±1.472) |
| boat2kit(MP-Nav Concept) | 0.275 | 0.041(±0.015) | 0.327(±0.055) | 0.610(±0.053) | 15.015(±1.217) |
| boat2dog(MP-Nav Instance) | 0.238 | 0.029(±0.021) | 0.324(±0.064) | 0.618(±0.023) | 14.952(±1.027) |
| boat2kit(MP-Nav Con.+Ins.) | 0.275 | 0.048(±0.008) | 0.418(±0.027) | 0.677(±0.035) | 14.602(±0.814) |

*Table 2.* Results of A2B attack on the COCO dataset. We report the median results with the standard deviations over 5 repeated training with different random seeds.

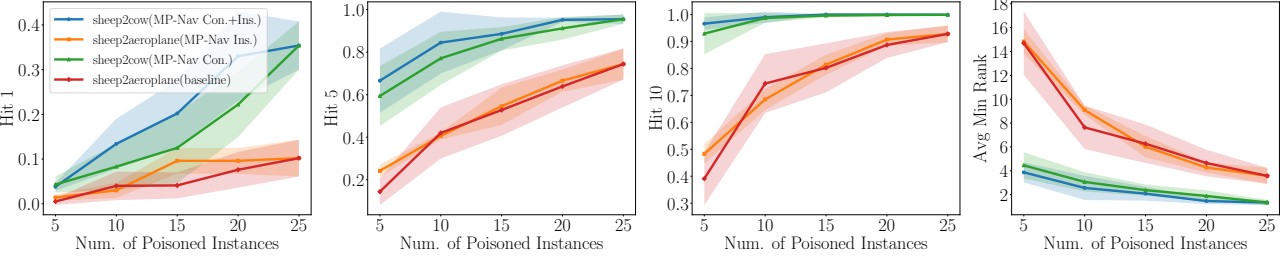

*Figure 2.* Results of A2B attack on the Flickr-PASCAL dataset. We report the median results with the standard deviations of various attacker budgets. We repeat the training over 5 trials with different random seeds.

Yang et al. (2023), we reported two metrics **Hit@K** and **MinRank** to evaluate the poisoning attacks in TIR task. Given a text descriptions of the $\mathcal{O}$, Hit@K measures the likelihood that the targeted image of concept $\mathcal{T}$ is retrieved within the top K-ranked results. It is calculated as the fraction of queries for which the images of $\mathcal{T}$ are included in the first K-ranked retrievals. A higher Hit@K value indicates a more effective poisoning attack. Given a text descriptions of the $\mathcal{O}$, MinRank reports the minimum ranking position of the targeted image $\mathcal{T}$ appearing among all retrieved images. A lower MinRank signifies that the targeted image of $\mathcal{T}$ is retrieved earlier, indicating a stronger attack.

**Learning Procedure** Following the previous work (Carlini & Terzis, 2022; Yang et al., 2023), we employ CLIP ViT-B/32 (Radford et al., 2021) as the backbone multimodal model, and conduct the poisoning attacks during the fine-tuning process. We fine-tuned the with 10 epochs with 128 batchsize using AdamW optimizer, initial learning rate of 0.2, cosine scheduler of 1.0 decay rate and weight decay of 0.2. Then, we evaluate the model utility and poisoning efficacy of the fine-tuned CLIP on COCO and Flickr-PASCAL dataset, repetitively.

**MP-Nav for Setting Selections** Yang et al. (2023) chose the default setting of boat2dog for COCO dataset and sheep2aeroplane for Flickr-PASCAL dataset, but the MP-Nav instead suggests using boat2kit and sheep2cow, respectively. Table 2 presents the experimental results of A2B attacks on the COCO dataset. The baseline method (Yang et al., 2023) selects the concepts "boat" and "dog" for misassociation, where 284 instances (out of 119,387 training instances) are randomly poisoned. In con-

trast, the MP-Nav firstly estimates the similarity scores of all concept pairs. The similarity score for the "boat" and "kit" pair is computed as 0.275, which is higher than that of the "boat" and "dog" pair (0.238). Consequently, MP-Nav identifies the "kit" as the better targeted concept, given the original concept "boat". Table 2 compares the baseline (Yang et al., 2023) with three MP-Nav configurations: Concept-level Selection, Instance-level Selection, and combined Concept+Instance selection. Across all configurations, MP-Nav significantly improves Hit@1, Hit@5, and Hit@10 (higher is better) and reduces Min Rank (lower is better). Notably, the combined Concept+Instance selection achieves the best results, with Hit@10 increasing from 0.505 to 0.677 and Min Rank reduced from 15.705 to 14.602.

Figures 2 present the experimental results of A2B attacks on the Flickr-PASCAL dataset. The baseline method (Yang et al., 2023) randomly selects the concepts "sheep" and "aeroplane" for misassociation. In contrast, MP-Nav calculates similarity scores based on the COCO dataset and determines that the similarity between "sheep" and "cow" (0.316) is higher than that between "sheep" and "aeroplane" (0.212). Consequently, MP-Nav prioritizes "sheep" and "cow" as the target pair for misassociation, then leveraging both instance-level selection to improve further attack success rates. Figures 2 plot the attack performance as attacker's budget increases. Across all metrics, MP-Nav (Con.+Ins.) consistently outperforms the baseline. Notably, at the 25 poisoned instances, the results of sheep2cow(MP-Nav Con.+Ins.) and sheep2cow(MP-Nav Con.) are the same, and results of sheep2aeroplane(baseline) are also the same as sheep2aeroplane(MP-Nav Ins.). This is because COCO training set contains only 25 images per concept.

Therefore, when the attacker's budget reaches 25 poisoned instances, the instance-level selection utilizes all available images, leading to identical results.

| Models | Flickr-PASCAL (1K test set) | | COCO (3.9K test set) | |
|---|---|---|---|---|
| | R@10(TR) | R@10(IR) | R@10(TR) | R@10(IR) |
| Clean | 98.5(±0.28) | 96.1(±0.13) | 93.0(±0.43) | 85.8(±0.53) |
| Baseline(Yang et al., 2023) | 99.0(±0.20) | 96.4(±0.17) | 92.9(±0.56) | 85.7(±0.46) |
| Base.+MP-Nav(Ins. Only) | 98.9(±0.14) | 96.5(±0.11) | 92.9(±0.46) | 85.7(±0.52) |
| MP-Nav(Con. Only) | 98.8(±0.22) | 96.4(±0.20) | 92.9(±0.43) | 85.8(±0.49) |
| MP-Nav(Con.+Ins.) | 99.0(±0.26) | 96.3(±0.16) | 93.1(±0.43) | 85.7(±0.41) |

*Table 3.* Comparison of model utility of CLIP models fine-tuned on clean and poisoned COCO and Flicker-PASCAL datasets. TR: Text Retrieval, IR: Image Retrieval. The poisoned models preserves the comparable utilities with the clean model.

Furthermore, we provided Table 3 that compare the utility scores of clean and poisoned CLIP fine-tunned on both Flicker-PASCAL and COCO datasets. A more comprehensive comparisons including R@1, R@5 scores are put into Table 6 in the Appendix. In Table 3, the clean model is the CLIP model trained on clean data without poisoning. The poisoned CLIP maintain the comparable results with the clean models.

In the Appendix, Figures 5 and 6 and Table 5 visualize a comprehensive result. The results suggest the MP-Nav's Concept-level and Instance-level selections cannot definitely improve data poisoning attacks. Many factors could contribute to the large uncertainties including learner's optimization procedures, data qualities and quantities, etc. However, the overall trend suggests attacker should at least follow the MP-Nav's suggested concepts and instances, to improve the chances of successful data poisoning attacks.

### 4.2. MP-Nav for ShadowCast Attack against VQA Task

In this section, we experimentally use MP-Nav to evaluate (or even improve) another multimodal data poisoning attack—ShadowCast (Xu et al., 2024).

**Clean Dataset for a VQA Task**  Following Xu et al. (2024), we used the clean MiniGPT4 dataset (Zhu et al., 2024) that consists of 3,500 detailed image description pairs for visual instruction tuning. Then, we crafted and injected a separate poisoned set into the clean MiniGPT4 set and used the combined dataset to fine-tune a multimodal model.

**Visual Instruction Tuning**  Following Xu et al. (2024), we utilized the opensourced LLaVA-1.5 (Liu et al., 2024) as the pre-trained model for visual instruction tuning, in which vision encoder is frozen and the language model with LoRA is trained. For each dataset, we trained LLaVA-1.5 model for 1 epochs using AdamW optimizer of learning rate $2e^{-4}$.

**Evaluation Metric**  For poisoning efficacy, we reported *attack success rate*. The targeted concept $\mathcal{T}$ is a class label,

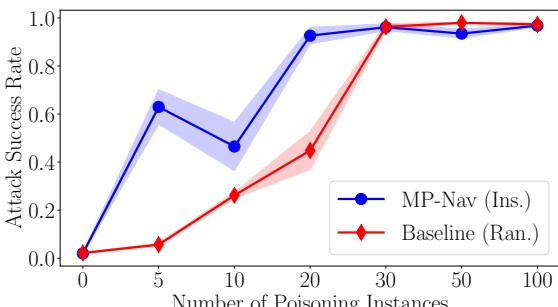

*Figure 3.* Results of ShadowCast attack on using Biden-Trump dataset. The learner can get access to 5 benign "Biden" instances, and the attacker can inject from 0 to 100 poisoning "Biden" instances. We report the mean results with the standard deviations of three evaluations prompts: "Who is person", "What is the name of this person" and "Identify the individual in this image".

we measure the attack success rate by the percentage of model responses on the test set that correctly mention $\mathcal{T}$ without mentioning the original concept $\mathcal{O}$.

For model utility, we reported the performance of clean and poisoned LLaVA-1.5 models on GQA benchmarks (Hudson & Manning, 2019). If we observe the utility of the poisoned model similar to the clean model, the attacks can primarily preserve the poisoned model's utility.

**MP-Nav for Poisoned Set Selections**  Xu et al. (2024) collected data from the Internet and curated image-text pairs of "Biden" and "Trump" concepts, with each concept comprising 277 training instances and 200 test instances. Using MP-Nav, we calculated the concept similarity to be 0.363, which is higher than other concept pairs such as "Biden-Hepburn" (0.210) (when we sourced additional instances of "Hepburn" and more). This relatively high similarity score indicates the effectiveness of the concept pairs and instances cherry-picked by Xu et al. (2024).

However, we found that benign "Biden" images can significantly dilute the effect of poisoned "Biden" images. Xu et al. (2024) injected the poisoning noises to all 277 training instances of "Biden" and assumed the absence of any benign "Biden" instances, which is a less practical setting. In practice, the attacker cannot control the entire training process and cannot guarantee that the learner has no access to clean "Biden" images. In contrast, we instead assume the learner can get access to some clean "Biden" images, and the attacker can manipulate only a subset of the 277 training instances. This highlights the importance of MP-Nav's instance-level selection component for effectively executing the ShadowCast attack.

In Figure 3, we conducted experiments on allowing attacker to manipulate n "Biden" images, where $n \in \{0, 5, 10, 20, 30, 50, 100\}$, and the learner fixed 5 clean

| Method | Clean | p=0.14% | p=0.27% | p=0.55% | p=0.82% | p=1.37% | p=2.74% | p=4.12% | p=5.49% |
|---|---|---|---|---|---|---|---|---|---|
| Baseline | 59.5(±0.07) | 59.6(±0.12) | 59.2(±0.06) | 59.7(±0.05) | 59.6(±0.09) | 59.5(±0.10) | 59.5(±0.09) | 59.4(±0.08) | 59.5(±0.07) |
| MP-Nav(Con. only) | 59.4(±0.04) | 59.5(±0.07) | 59.1(±0.10) | 59.7(±0.04) | 59.4(±0.09) | 59.4(±0.08) | 59.4(±0.06) | 59.5(±0.05) | 59.6(±0.04) |
| MP-Nav(Ins. only) | 59.5(±0.08) | 59.8(±0.08) | 59.2(±0.07) | 59.7(±0.08) | 59.6(±0.12) | 59.6(±0.13) | 59.5(±0.14) | 59.5(±0.08) | 59.8(±0.22) |
| MP-Nav(Con.+Ins.) | 59.5(±0.07) | 59.5(±0.06) | 59.0(±0.08) | 59.5(±0.07) | 59.4(±0.10) | 59.5(±0.05) | 59.4(±0.05) | 59.6(±0.04) | 59.5(±0.04) |

*Table 4.* Performance of clean and poisoned LLaVA-1.5 models on GQA benchmark (the higher, the better). p denotes the proportion of poison samples in Food101 dataset. This shows the poisoned models preserves the comparable model utilities with the clean model.

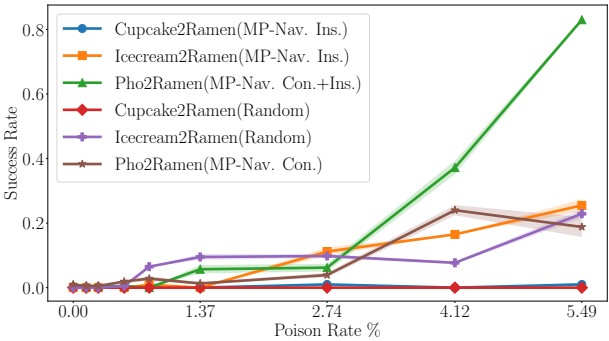

*Figure 4.* Results of ShawdowCast attack on using Food101 dataset. We report the median results with the standard deviations of various attacker budgets. We repeat the training over 5 trials with different random seeds.

"Biden" images. Then, we combined the poisoned set with the clean "Trump" instances and the MiniGPT4 set for the instruction tuning. Figure 3 shows under the small poisoning budget (such as $\eta = (0, 20]$, MP-Nav's instance-level component can significantly enhance the efficacy of ShadowCast attacks. In the Appendix, we illustrate a more comprehensive result when the learner can get access to more benign "Biden" instances.

Besides, to conduct control experiments, we leverage the open-sourced Food101 dataset (Bossard et al., 2014) that consists of 101 food categories with 750 training and 250 test images per category, making a total of 101k images. Given each image and its corresponding label, we generate the caption descriptions using the opensourced LLaVA-v1.5-7b model (Liu et al., 2024). Then, we can use MP-Nav to select concept pairs and instances, and make controlled comparisons of Shadowcast performance. Note that in ShadowCast, the attacker adds the PGD noise to instances of targeted concept. Then, we fixed concept "ramen" as the targeted concept. MP-Nav suggested "pho" as the source concept because the similarity is highest (0.268). We compare two randomly chosen concept, e.g. "cupcake" and "icecream" (similarity scores are 0.126 and 0.176, respectively). Again, MP-Nav can effectively help ShadowCast attacks on Food101 dataset.

## 5. Conclusion

This paper has introduced a plug-and-play module MP-Nav that is designed to evaluate data poisoning attacks against multimodal learning models. By leveraging concept-level and instance-level selection components, MP-Nav has identified semantically similar concept pairs and robust instances to help attackers to craft better poisoned instances. Experimental results have demonstrated its effectiveness in the existing state-of-the-art poisoning attacks (such as AtoB and ShadowCast) across different multimodal tasks such as Text-Image Retrieval (TIR) and Visual Question Answering (VQA). Additionally, MP-Nav can also preserve the utility of poisoned models and ensure that the attacks remain stealthy. Our findings underscore the vulnerabilities in multimodal models, calling for the development of robust defenses to counteract the data poisoning threats.

## Impact Statement

MP-Nav exhibits the dual-use nature. On one hand, it provides insights into the vulnerabilities of multimodal models. On the other hand, it raises concerns in real-world scenarios. The ability to easily and stealthily manipulate multimodal models could be exploited by malicious parties to propagate disinformation, manipulate public perception, or even disrupt critical AI-powered systems, such as healthcare diagnostics, autonomous vehicles, and public safety infrastructures. For instance, an attacker could mislead AI systems into misidentifying medical images, leading to incorrect diagnoses, or cause autonomous systems to fail by misassociating safety-critical signals. Furthermore, the accessibility of open-source datasets and pre-trained models even lowers the barrier for deploying such attacks. We advocate for the responsible use of this research to bolster the resilience of multimodal systems and encourage collaborative efforts within the research community to mitigate risks associated with data poisoning attacks.

## Acknowledgment

This paper is supported in part by the Army Research Office under grant number W911NF-21-1-0155 and by the New York University Abu Dhabi (NYUAD) Center for Artificial Intelligence and Robotics, funded by Tamkeen under the NYUAD Research Institute Award CG010.

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

# A. Extended Literature Review

### A.1. Multimodal Learning

Multimodal learning, which integrates data from multiple modalities such as vision, text, audio, and sensory inputs, has been a cornerstone of artificial intelligence research. Early efforts in the 1990s primarily focused on sensor fusion, combining multiple sources of information to improve decision-making accuracy (Hall & Llinas, 1997).

With the rise of deep learning, Ngiam et al. (2011) demonstrated that deep neural networks could effectively fuse multimodal features, enhancing representations of one modality when additional modalities were included. This foundational work highlighted the potential of leveraging cross-modal information for feature learning.

The success of convolutional neural networks (CNNs) following the ImageNet competition in 2012 (Krizhevsky et al., 2012) further spurred research into multimodal fusion techniques, particularly between visual and other modalities. Antol et al. (2015) extended these efforts with the development of Visual Question Answering (VQA), a system that integrated CNNs for image understanding with recurrent neural networks (RNNs) for natural language processing.

The introduction of transformers (Vaswani, 2017) in 2017 represented a paradigm shift. Initially designed for sequence modeling, transformers quickly became the backbone of multimodal systems due to their ability to process large-scale data efficiently. Models such as CLIP (Radford et al., 2021), BLIP (Li et al., 2022), and ALBEF (Li et al., 2021) exemplify the application of transformer-based architectures to align and fuse vision and language representations, achieving state-of-the-art performance in various multimodal tasks.

### A.2. Multimodal Contrastive Learning

Multimodal contrastive learning has emerged as a powerful approach to aligning data from different modalities by projecting them into a shared embedding space. The core idea is to maximize the similarity of positive sample pairs (e.g., semantically related image-text pairs) while minimizing the similarity of negative sample pairs (e.g., unrelated image-text pairs). Similarity is commonly measured using cosine similarity.

Representative models like CLIP (Radford et al., 2021) and ALIGN (Jia et al., 2021) have demonstrated the efficacy of contrastive learning by aligning large-scale image-text pairs in a shared feature space. These models significantly improve zero-shot learning capabilities, enabling tasks such as image classification and retrieval without task-specific annotations. By aligning features in a balanced way, multimodal contrastive learning avoids issues of information imbalance across modalities, which often arise when combining feature vectors of varying dimensions.

### A.3. Multimodal Text-Image Retrieval Task

Cross-modal image-text retrieval (ITR) is a fundamental task in multimodal learning, with applications in search engines, content recommendation, and more. ITR typically includes two subtasks: image-to-text (i2t) and text-to-image (t2i) retrieval (Cao et al., 2022).

Given a model and an input text or image, the model maps the inputs into embedded vectors in a shared semantic space. The goal is to retrieve data from the other modality that maximizes the similarity. Advances in ITR have been driven by contrastive learning techniques (Radford et al., 2021; Li et al., 2021; Jia et al., 2021), which align positive sample pairs and separate negative pairs to improve retrieval accuracy.

### A.4. Review of Poisoning Attacks

Poisoning attacks aim to manipulate a model by injecting adversarial data into the training set. Formally, given a clean dataset $D_c = \{(x_i, y_i)\}$, an attacker constructs a poisoned dataset $D_p = \{(x_i', y_i')\}$ and combines it with the clean data to form $D_{train} = D_c \cup D_p$. The goal is to alter the model's behavior for specific inputs while maintaining overall performance.

# B. Extended Experiment

Table 5 shows the comprehensive experiments results of AtoB attack using COCO dataset.

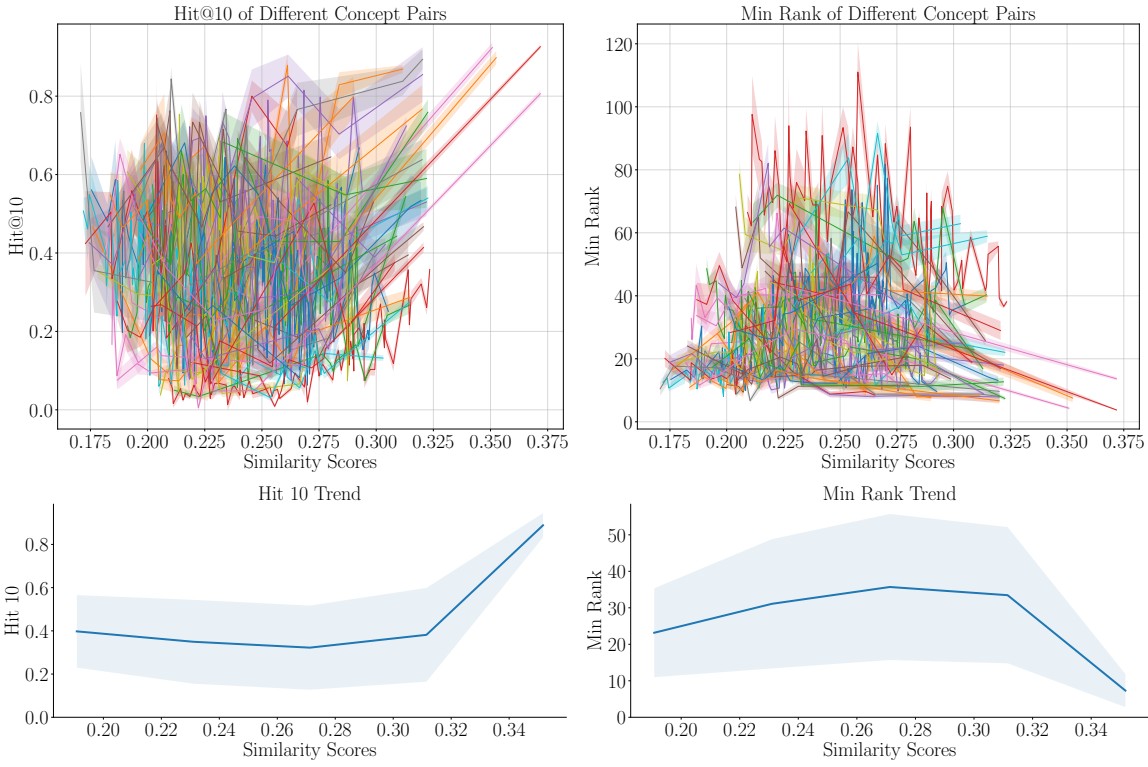

*Figure 5.* A comprehensive comparison of different concept pairs of various similarity scores for the COCO dataset. In the upper two plots, each line represents a fixed source concepts with different targeted concepts (of different similarity scores): The average results and standard deviation (in shade) over repeated trials are reported. In the bottom two plots, we buckets the similarity scores over 6 bins and average results to visualize the trend.

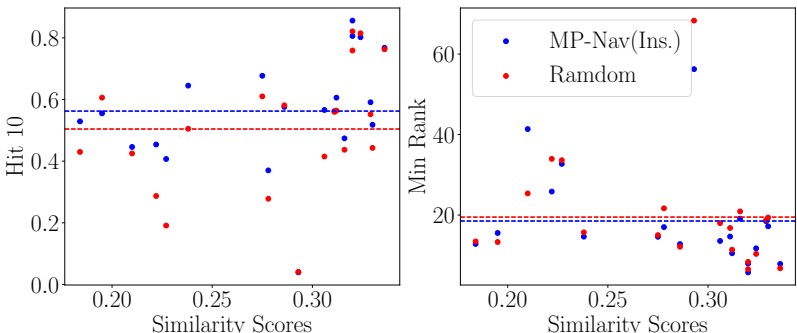

*Figure 6.* Scatter plots of the Table 5 for a better visualization. Each dashed line denotes the average performance of each instance selection strategy (i.g., MP-Nav Instance-level selection and baseline's random selection.

| Source | Target | Similarity | Attack Method | Hit@1 ↑ | Hit@5 ↑ | Hit@10 ↑ | Average MinRank ↓ |
|---|---|---|---|---|---|---|---|
| banana | skis | 0.210 | Baseline (Random) | 0.006(±0.005) | 0.160(±0.057) | 0.425(±0.095) | 25.369(±3.807) |
|  | apple | 0.324 | MP-Nav Concept | 0.101(±0.015) | 0.582(±0.051) | 0.815(±0.040) | 10.350(±1.439) |
|  | skis | 0.210 | MP-Nav Instance | 0.022(±0.011) | 0.214(±0.033) | 0.446(±0.034) | 41.360(±3.276) |
|  | apple | 0.324 | MP-Nav Con.+Ins. | 0.128(±0.012) | 0.586(±0.024) | 0.778(±0.017) | 12.205(±1.121) |
| broccoli | bird | 0.222 | Baseline (Random) | 0.004(±0.004) | 0.098(±0.047) | 0.287(±0.083) | 33.958(±5.365) |
|  | carrot | 0.336 | MP-Nav Concept | 0.080(±0.019) | 0.511(±0.037) | 0.763(±0.032) | 6.774(±0.608) |
|  | bird | 0.222 | MP-Nav Instance | 0.012(±0.009) | 0.230(±0.028) | 0.448(±0.028) | 31.342(±2.123) |
|  | carrot | 0.336 | MP-Nav Con.+Ins. | 0.076(±0.015) | 0.493(±0.043) | 0.730(±0.021) | 8.602(±1.037) |
| bus | bird | 0.227 | Baseline (Random) | 0.001(±0.003) | 0.070(±0.033) | 0.191(±0.061) | 33.617(±3.387) |
|  | car | 0.3105 | MP-Nav Concept | 0.041(±0.014) | 0.280(±0.037) | 0.560(±0.046) | 16.785(±1.262) |
|  | bird | 0.227 | MP-Nav Instance | 0.012(±0.014) | 0.184(±0.112) | 0.407(±0.129) | 32.709(±4.543) |
|  | car | 0.3105 | MP-Nav Con.+Ins. | 0.070(±0.020) | 0.318(±0.043) | 0.562(±0.031) | 14.658(±0.487) |
| fork | carrot | 0.316 | Baseline (Random) | 0.044(±0.011) | 0.228(±0.021) | 0.437(±0.036) | 20.902(±1.625) |
|  | spoon | 0.330 | MP-Nav Concept | 0.057(±0.013) | 0.256(±0.026) | 0.443(±0.028) | 19.382(±1.417) |
|  | carrot | 0.316 | MP-Nav Instance | 0.038(±0.009) | 0.280(±0.023) | 0.500(±0.022) | 20.570(±1.727) |
|  | spoon | 0.330 | MP-Nav Con.+Ins. | 0.048(±0.017) | 0.267(±0.024) | 0.471(±0.031) | 18.665(±1.545) |
| giraffe | carrot | 0.184 | Baseline (Random) | 0.007(±0.006) | 0.165(±0.059) | 0.430(±0.081) | 13.454(±1.613) |
|  | zebra | 0.320 | MP-Nav Concept | 0.039(±0.015) | 0.447(±0.086) | 0.759(±0.087) | 6.535(±1.280) |
|  | carrot | 0.184 | MP-Nav Instance | 0.010(±0.011) | 0.167(±0.060) | 0.529(±0.073) | 12.820(±1.692) |
|  | zebra | 0.320 | MP-Nav Con.+Ins. | 0.042(±0.009) | 0.513(±0.071) | 0.820(±0.071) | 6.104(±0.954) |
| pizza | cake | 0.278 | Baseline (Random) | 0.009(±0.006) | 0.124(±0.026) | 0.278(±0.053) | 21.672(±2.769) |
|  | fork | 0.312 | MP-Nav Concept | 0.036(±0.011) | 0.310(±0.034) | 0.564(±0.037) | 11.374(±1.056) |
|  | cake | 0.278 | MP-Nav Instance | 0.013(±0.009) | 0.170(±0.015) | 0.370(±0.035) | 17.002(±1.024) |
|  | fork | 0.312 | MP-Nav Con.+Ins. | 0.052(±0.008) | 0.343(±0.018) | 0.606(±0.039) | 10.508(±0.972) |
| sheep | snowboard | 0.195 | Baseline (Random) | 0.022(±0.013) | 0.302(±0.081) | 0.606(±0.101) | 13.300(±2.076) |
|  | cow | 0.320 | MP-Nav Concept | 0.084(±0.031) | 0.560(±0.068) | 0.821(±0.067) | 8.358(±1.009) |
|  | snowboard | 0.195 | MP-Nav Instance | 0.013(±0.009) | 0.254(±0.075) | 0.555(±0.101) | 15.546(±2.594) |
|  | cow | 0.320 | MP-Nav Con.+Ins. | 0.093(±0.015) | 0.559(±0.037) | 0.830(±0.023) | 8.300(±0.562) |
| skateboard | backpack | 0.286 | Baseline (Random) | 0.040(±0.017) | 0.324(±0.049) | 0.581(±0.048) | 12.126(±1.323) |
|  | person | 0.293 | MP-Nav Concept | 0.002(±0.003) | 0.015(±0.006) | 0.041(±0.012) | 68.315(±7.646) |
|  | backpack | 0.286 | MP-Nav Instance | 0.027(±0.008) | 0.284(±0.044) | 0.576(±0.037) | 12.789(±1.010) |
|  | person | 0.293 | MP-Nav Con.+Ins. | 0.001(±0.002) | 0.013(±0.006) | 0.039(±0.005) | 56.249(±3.598) |
| traffic light | truck | 0.306 | Baseline (Random) | 0.022(±0.011) | 0.225(±0.033) | 0.415(±0.034) | 17.990(±1.725) |
|  | car | 0.329 | MP-Nav Concept | 0.063(±0.018) | 0.381(±0.029) | 0.552(±0.024) | 18.770(±1.385) |
|  | truck | 0.306 | MP-Nav Instance | 0.034(±0.013) | 0.351(±0.048) | 0.566(±0.044) | 13.563(±1.095) |
|  | car | 0.329 | MP-Nav Con.+Ins. | 0.045(±0.011) | 0.374(±0.031) | 0.571(±0.029) | 19.437(±2.201) |

*Table 5.* More results of A2B attack on the COCO dataset in different concept pairs.

| Method | Flickr-PASCAL (1K test set) | | | | | | COCO (3.9K test set) | | | | | |
|---|---|---|---|---|---|---|---|---|---|---|---|---|
| | TR | | | IR | | | TR | | | IR | | |
| | R@1 | R@5 | R@10 | R@1 | R@5 | R@10 | R@1 | R@5 | R@10 | R@1 | R@5 | R@10 |
| Clean | 86.4(±0.70) | 96.6(±0.14) | 98.5(±0.28) | 73.7(±0.45) | 92.8(±0.16) | 96.1(±0.13) | 63.4(±0.67) | 86.8(±0.68) | 93.0(±0.43) | 48.4(±0.88) | 76.6(±0.65) | 85.8(±0.53) |
| Baseline | 86.8(±0.66) | 97.4(±0.32) | 99.0(±0.20) | 73.7(±0.39) | 92.8(±0.20) | 96.4(±0.17) | 63.5(±0.94) | 86.7(±0.63) | 92.9(±0.56) | 48.2(±0.81) | 76.7(±0.64) | 85.7(±0.46) |
| Base.+MP-Nav(Ins. Only) | 87.5(±0.46) | 97.3(±0.36) | 98.9(±0.14) | 74.0(±0.29) | 93.1(±0.27) | 96.5(±0.11) | 63.8(±0.73) | 87.0(±0.64) | 92.9(±0.46) | 48.3(±0.95) | 76.6(±0.62) | 85.7(±0.52) |
| MP-Nav(Con. Only) | 87.0(±0.60) | 97.2(±0.23) | 98.8(±0.22) | 73.7(±0.43) | 92.8(±0.25) | 96.4(±0.20) | 63.6(±0.98) | 86.6(±0.66) | 92.9(±0.43) | 48.3(±0.85) | 76.6(±0.68) | 85.8(±0.49) |
| MP-Nav(Con.+Ins.) | 86.9(±0.33) | 97.2(±0.43) | 99.0(±0.26) | 73.9(±0.43) | 92.8(±0.34) | 96.3(±0.16) | 63.3(±0.70) | 86.9(±0.59) | 93.1(±0.43) | 48.2(±0.63) | 76.7(±0.54) | 85.7(±0.41) |

*Table 6.* Comparison of model utilities clean and poisoned CLIP fine-tuned on Flickr-PASCAL and COCO datasets. TR: Text Retrieval, IR: Image Retrieval.

