# OpenReview forum: "MP-Nav: Enhancing Data Poisoning Attacks against Multimodal Learning"
_ICML.cc/2025/Conference — ICML 2025 poster_

### Official Review · Reviewer_CBYT · 2025-02-19

**Overall Recommendation:** 3

**Summary:**

1. The author analyzed the shortcomings of existing attack methods: only associating errors by randomly selecting concepts, and poisoning instances randomly, which usually makes it difficult to achieve a good attack effect.
2. The authors proposed a plug-and-play module MP-Nav. MP-Nav effectively solves the problems existing in the random selection method of existing methods by identifying semantically similar concepts at both the concept and instance levels and selecting robust instances, and effectively improves the attack effect.
3. Experiments have confirmed that MP-Nav can significantly enhance the effectiveness of the most advanced data poisoning attacks, i.e., AtoB and ShadowCast in multimodal tasks, and maintain the practicality of the model in various datasets.

**Claims And Evidence:**

Although the author pointed out the problem that existing methods can hardly achieve good attack effects only by randomly selecting concepts, the author only showed statistical experimental results, and did not have specific visualization experiments to prove their view point.

**Essential References Not Discussed:**

There are only two Baseline methods in the experimental part: AtoB and ShadowCast. This makes it difficult to fully prove the applicability of the proposed plug-and-play MP-Nav. The authors should consider using some other methods as baselines, such as [1][2][3].

[1] Data Poisoning Attacks Against Multimodal Encoders, Ziqing Yang, Xinlei He, Zheng Li, M. Backes, Mathias Humbert, Pascal Berrang, Yang Zhang, International Conference on Machine Learning, 2022.
[2] CleanCLIP: Mitigating Data Poisoning Attacks in Multimodal Contrastive Learning, Hritik Bansal, Nishad Singhi, Yu Yang, Fan Yin, Aditya Grover, Kai-Wei Chang, IEEE International Conference on Computer Vision, 2023.
[3] Backdooring Multimodal Learning, Xingshuo Han, Yutong Wu, Qingjie Zhang, Yuan Zhou, Yuan Xu, Han Qiu, Guowen Xu, Tianwei Zhang, IEEE Symposium on Security and Privacy, 2024.

**Experimental Designs Or Analyses:**

Although the author conducted a large number of attack experiments to verify the effectiveness of the proposed method, they used too few baseline methods. There are only two baseline methods in the experimental part: AtoB and ShadowCast, which makes it difficult to fully prove the applicability of the proposed plug-and-play MP-Nav. The author should consider using some other methods as baselines, e.g., [1-3].

[1] Data Poisoning Attacks Against Multimodal Encoders, Ziqing Yang, Xinlei He, Zheng Li, M. Backes, Mathias Humbert, Pascal Berrang, Yang Zhang, International Conference on Machine Learning, 2022.
[2] CleanCLIP: Mitigating Data Poisoning Attacks in Multimodal Contrastive Learning, Hritik Bansal, Nishad Singhi, Yu Yang, Fan Yin, Aditya Grover, Kai-Wei Chang, IEEE International Conference on Computer Vision, 2023.
[3] Backdooring Multimodal Learning, Xingshuo Han, Yutong Wu, Qingjie Zhang, Yuan Zhou, Yuan Xu, Han Qiu, Guowen Xu, Tianwei Zhang, IEEE Symposium on Security and Privacy, 2024.

**Methods And Evaluation Criteria:**

yes

**Other Comments Or Suggestions:**

Please answer all my doubts carefully, and I will adjust my final score based on the answers.

**Other Strengths And Weaknesses:**

Strengths:
This paper summarizes the key problems of existing multi-modal poisoning attack methods, namely they only associate concepts randomly and poison instances randomly, which usually makes it difficult to achieve a good attack effect.
2. The author made improvements to the above issues and proposed a plug-and-play module MP-Nav, which includes two components: Concept-level Selection and Instance-level Selection.
3. The author conducted a large number of experiments to verify the effectiveness of their method.

Weaknesses:
1. The author didn't conduct a detailed experimental analysis and explanation of the essential reasons for the existing method problems, so readers may find it difficult to deeply understand the starting point of this article.
2. Table 2 and Figure 2 represent the effectiveness of the author's method on two different datasets. However, on the PASCAL dataset, the effect of Instance-level Selection is negligible, and the author should explain why the results are inconsistent with those on the COCO dataset.
3. Since the proposed method is a plug-and-play module, the author should choose more methods as baselines.

**Questions For Authors:**

1. How is the number of concept centers determined, and is it related to the number of instance categories?
2. In lines 232 to 234, are the images and texts of the original concept pairs (I_A, T_A) and the target concept pairs (I_B, T_B) being swapped? After the swap, does the original concept pair become (I_A, T_B), and the target concept pair become (I_B, T_A)?

**Relation To Broader Scientific Literature:**

The method proposed in this paper is a plug-and-play module, and the author has verified its effectiveness on the baselines method, i.e., AtoB and ShadowCast.

**Theoretical Claims:**

The author conducted experimental verification of the proposed method, but lacked a detailed experimental analysis of the shortcomings of existing methods.

---

> ### Author Rebuttal · Authors · 2025-03-31
>
> Thanks for your positive score. Please find our responses below.
>
> 1 [Essential References Not Discussed]: “The author should consider using some other methods as baselines [1-3]”\
> **Response** 1:
> We have indeed used [1] as one of the baseline methods that our paper has made comparisons with.
> [2] focuses on the backdoor defense (and not poisoning attack) of CLIP, and [3] focuses on the backdoor attacks for multimodal learning.
> Table 1 in our paper has highlighted the setting differences of data poisoning and backdoor triggers.
> Nevertheless, MP-Nav is a plug-and-play module, which can potentially enhance other types of attacks, including backdoor attacks, adversarial evasion attacks, model inversion attacks, etc. In revision, we will cite the relevant papers [1-3], and in the future, we will compare MP-Nav with [3] and produce more results.
>
> 2 [Other Strengths And Weaknesses]: “The author didn't conduct a detailed experimental analysis and explanation of the essential reasons for the existing method problems, so readers may find it difficult to deeply understand the starting point of this article.”\
> **Response** 2:
> The two methods (existing random selection method and our MP-Nav method) fundamentally differ in strategy: random selection relies on stochastic trials, but MP-Nav relies on principled guidance to enhance poisoning efficacy. Specifically, random selections frequently choose concept pairs that are semantically distant or instances that poorly represent the targeted concepts, resulting in easily diluted poisoning effects by benign instances.
> In revision, we plan to add a visualization experiment on showing the above fact. Since rebuttal is character-limited only, kindly allow us to describe this visualization experiment: We will use PCA/T-SNE to reduce high-dimensional features into 2-D vectors and visualize and compare differences between embeddings of poisoned and benign instances. We will also plot the embedding evolutions over the training epochs and compare how MP-Nav selected poisoning instances differ from randomly selected poisoning instances, w.r.t. the counterpart benign instances.
>
> 3 [Other Strengths And Weaknesses]: “Table 2 and Figure 2 represent the effectiveness of the author's method on two different datasets. However, on the PASCAL dataset, the effect of Instance-level Selection is negligible, and the author should explain why the results are inconsistent with those on the COCO dataset.”\
> **Response** 3: In Table 2 of COCO dataset, we fixed the poisoning data of 284 (out of 119387 training data) and made fair comparisons with baseline A2B attack, which keeps the same as the baseline paper.
>
> However, in Figure 2 (PASCAL-Flicker combined dataset), the scenario differs. Flicker is large image-text pairs (without ground truth concepts) of 29000 images, where the attacker does not touch.
> The PASCAL dataset contains only 500 labeled images (for training) divided equally across 20 concepts, leaving a maximum of 25 instances per concept available for poisoning. In the original baseline paper, all 25 instances per concept were poisoned—an impractical scenario in real-world settings. To reflect more realistic conditions, we reduced the attacker’s budget (number of allowed poisoned instances), poisoning fewer than the maximum number of available instances per concept, leaving the remainder as benign data. As a consequence, when the attacker’s budget reaches 25 poisoned instances, the MP-Nav instance-level selection and random selection both utilize all available instances, naturally leading to **identical** performance.
> Thus, the negligible difference at instance-level selection arises directly from the dataset limitation rather than inconsistency in MP-Nav’s effectiveness.
>
> 4 [Questions For Authors]: “How is the number of concept centers determined, and is it related to the number of instance categories?”\
> **Response** 4: Yes. Each category has one concept center. The concept center considers both image and text embeddings of the same category (concept).
>
> 5 [Questions For Authors]: “In lines 232 to 234, are the images and texts of the original concept pairs (I_A, T_A) and the target concept pairs (I_B, T_B) being swapped? After the swap, does the original concept pair become (I_A, T_B), and the target concept pair become (I_B, T_A)?”\
> **Response** 5: No, it is not being swapped. To make the fair comparison with baseline A2B attack [1], we follow the setting of [1], i.e., original concept pairs (I_A, T_A) and the target concept pairs (I_B, T_B)  -> (I_A, T_B) and (I_B, T_B) . Attackers only change part of texts of original concept instances, and do not touch target concept instances. Kindly refer to Section 2 Preliminary for details. \
> [1]  Data poisoning attacks against multimodal encoders, in ICML 2023

---

> > ### Comment · Reviewer_CBYT · 2025-04-03
> >
> > I confirm that I have read the author's response, and maintain my original score.

---

> > > ### Author Response · Authors · 2025-04-07
> > >
> > > Thank you for carefully reading our response.

---

### Official Review · Reviewer_Nd9w · 2025-03-11

**Overall Recommendation:** 4

**Summary:**

This paper introduces the Multimodal Poison Navigator (MP-Nav), a plug-and-play module designed to improve data poisoning attacks on multi-modal models. The authors propose a two-step approach: (1) concept-level selection, which identifies semantically similar concepts for misassociation, and (2) instance-level selection, which ranks and selects robust instances to maximize attack efficacy. The proposed method enhances existing data poisoning attacks such as AtoB and ShadowCast and is evaluated on Text-Image Retrieval (TIR) and Visual Question Answering (VQA) tasks. Experimental results show that MP-Nav significantly improves attack effectiveness while preserving model utility.

## update after rebuttal
The authors address most of my concerns. So I raised my score.

**Claims And Evidence:**

Yes, it is clear and convincing. The paper claims that MP-Nav enhances the efficacy of data poisoning attacks against multimodal models while maintaining model utility. The evidence provided includes empirical results on benchmark datasets (such as COCO, Flickr, PASCAL, and Food101), and customized “Biden-Trump”. Dataset. The evaluations demonstrate that MP-Nav improves attack success rates while keeping retrieval and classification performance intact.

**Essential References Not Discussed:**

All highly related works are discussed as far as I know.

**Experimental Designs Or Analyses:**

The experimental design follows the standard benchmarks and datasets. The comparison between baseline attacks and MP-Nav-enhanced attacks demonstrates meaningful improvements. Furthermore, in Figure 5 & Table 5 of the appendix, the paper has provided comprehensive evaluations of differently-similar concepts on the attack efficacy. This could confirm the observed performance gains.

**Methods And Evaluation Criteria:**

Yes, the proposed method makes sense. The evaluation criteria (such as Hit@K and MinRank for TIR and  attack success rate for VQA) are appropriate.

**Other Comments Or Suggestions:**

No further comments. Please see my questions below.

**Other Strengths And Weaknesses:**

This paper presents an effective enhancement to existing multimodal poisoning attacks. The modularity of MP-Nav allows for easy integration with various multimodal tasks. However, a discussion on potential countermeasures would add more depth.

**Questions For Authors:**

- Would MP-Nav induce the computational overhead for the attacker? Could you clarify the attacker (using MP-Nav) could be different from the attacker (using random selections)?
- Can MP-Nav be adapted to other attack paradigms beyond data poisoning?
- In Section 4.2, authors poisoned the LLaVA-1.5 model by fine-tuning it for 1 epoch. I understand this setting is originally from ShawdowCast paper. How about fine-tuning LLaVA for several epochs? Would the poison effect still stay under the presence of benign data?

**Relation To Broader Scientific Literature:**

The paper is well-positioned in the field of data poisoning attacks and multimodal learning.
It builds upon prior work in data poisoning but distinguishes itself by proposing a structured poisoning strategy tailored to different multimodal models.

**Theoretical Claims:**

Not applicable. This paper does not have theoretical claims.

---

> ### Author Rebuttal · Authors · 2025-03-31
>
> Thanks for your positive score. Please find our responses below.
>
> 1 [Other Strengths And Weaknesses]: “a discussion on potential countermeasures would add more depth.”\
> **Response** 1: This is a similar question to one raised by reviewer cLuX. Kindly refer to the "Response 3" for reviewer cLuX.
>
>
> 2 [Questions For Authors]: “Would MP-Nav induce the computational overhead for the attacker? Could you clarify the attacker (using MP-Nav) could be different from the attacker (using random selections)?”\
> **Response** 2: MP-Nav indeed introduces some computational overhead compared to a purely random selection strategy due to the computation of semantic embeddings of instances. MP-Nav requires o(nW) computation overhead for the model inference, where n refers to size of training set and W refers to size of open-sourced model parameters. In our experiment, a single 4090 GPU is sufficiently powerful for MP-Nav computations.
> Kindly note that the two methods (i.e., MP-Nav and random selections) fundamentally differ in strategy: random selection relies on stochastic trials, but MP-Nav relies on principled guidance to enhance poisoning efficacy.
>
>
> 3 [Questions For Authors]: “Can MP-Nav be adapted to other attack paradigms beyond data poisoning?”\
> **Response** 3: Yes. MP-Nav is a plug-and-play module, and we presume it can also enhance adversarial evasion attack and model extraction attacks. In evasion attacks, attackers could identify vulnerable pairs of concepts where decision boundaries are naturally close, thus focusing adversarial perturbations on the most vulnerable concept pairs.
> In model extraction attacks, MP-Nav’s embedding-based selection could assist attackers in choosing query samples that maximize information gain about internal model decision boundaries, facilitating faster or more effective extraction of the victim models. We will make a thorough exploration of the above attacks in future.
>
>
> 4 [Questions For Authors]: “In Section 4.2, authors poisoned the LLaVA-1.5 model by fine-tuning it for 1 epoch. I understand this setting is originally from ShawdowCast paper. How about fine-tuning LLaVA for several epochs? Would the poison effect still stay under the presence of benign data?”\
> **Response** 4: Yes. The poison attack is still effective under the presence of benign data. We have conducted additional experiments by extending fine-tuning over 4 epochs, and we report attack success rate (SR) in the table below. Interestingly, we have observed that benign data won’t dilute the poisoning effect over epochs, and the poisoning effect is further enhanced, especially under small poisoning ratio and when employing the MP-Nav’s robust instance selection. One plausible explanation is that large models tend to overfit specific associations over prolonged fine-tuning, thereby reinforcing the malicious associations introduced by carefully selected poisoned instances. We will add the additional results in revision.
> |Attack |Poison ratio | SR(Epoch #1) | SR(Epoch #2)| SR(Epoch #3)| SR(Epoch #4)|
> | --------| -------- | ------- |------- |------- |------- |
> | MP-Nav| 1% | 0.01 | 0.08| 0.58| 0.56|
> | Random| 1% | 0.01 | 0.02| 0.17| 0.23|
> | MP-Nav| 3% | 0.02 | 0.98| 0.97| 0.97|
> | Random| 3% | 0.01 | 0.62| 0.90| 0.89|

---

> > ### Comment · Reviewer_Nd9w · 2025-04-05
> >
> > Thanks for the clear responses and solid additional experiments on LLaVA-1.5 fine-tuning. The results demonstrating MP-Nav’s robustness and its ability to enhance poisoning efficacy even under the presence of benign data are compelling, and these findings significantly strengthen the paper. MP-Nav’s effectiveness and adaptability stand out. I’m happy to increase my score—great work!

---

> > > ### Author Response · Authors · 2025-04-07
> > >
> > > Thank you for carefully reading our response.

---

### Official Review · Reviewer_cLuX · 2025-03-12

**Overall Recommendation:** 4

**Summary:**

This paper presents MP-Nav that optimizes data poisoning attacks for vision-language models. The approach strategically selects concept pairs and robust instances to maximize poisoning efficiency while maintaining overall model utility. The authors evaluate MP-Nav on benchmark datasets and demonstrate improvements over existing poisoning attacks.

**Claims And Evidence:**

The claims regarding improved attack success are well-supported by extensive experiments. However, the claim that MP-Nav preserves model utility requires further clarification, as the poisoned data could have long-term effects on model behavior.

**Essential References Not Discussed:**

To the best of my knowledge, all relevant related works that should be mentioned have been included.

**Experimental Designs Or Analyses:**

The experimental setup is rigorous and the comparisons with baseline methods are fair. Thus, the experimental results are trustworthy.

**Methods And Evaluation Criteria:**

This paper mathematically formulates the proposed MP-Nav., which greatly helps readability. The proposed algorithm is clear and well correlated to Figure 1, so I can understand the method easily.

**Other Comments Or Suggestions:**

1 Please explicitly discuss the limitations of MP-Nav, particularly regarding scenarios where poisoning may not be effective.

**Other Strengths And Weaknesses:**

+Strengths: Clear motivation, comprehensive experimental results, and a well-structured poisoning approach.
-Weaknesses: Limited exploration of countermeasures and lack of explicit discussion of the limitation of the proposed MP-Nav.

**Questions For Authors:**

1 Have authors observed any situations where MP-Nav fails to improve poisoning efficacy, or where it inadvertently lowers the effectiveness of an attack compared to baseline methods?
2 From my understanding, MP-Nav currently assumes access to embeddings for similarity computation. How well does it perform if the attacker does not have access to the model’s internal representations?
3 Could existing outlier detection or data sanitization methods be used to identify and filter poisoned samples before model training?

**Relation To Broader Scientific Literature:**

How are the key contributions of the paper related to the broader scientific literature? Be specific in terms of prior related findings/results/ideas/etc.
The paper contributes to adversarial machine learning research. The prior studies neglect the importance of concepts and instance-level selections for effectively evaluating the threat of data poisoning attacks. With this regard, this paper has made the remedy by proposing an MP-Nav method that can comprehensively enhance multi-modal data poisoning attacks.

**Theoretical Claims:**

The methods are clearly described, and the evaluation criteria align well with the research objectives. However, the paper lacks an explicit discussion on the limitations of MP-Nav, particularly regarding scenarios where poisoning may not be effective.

---

> ### Author Rebuttal · Authors · 2025-03-31
>
> Thanks for your positive review. Please find our responses below.
>
> 1 [Other Comments Or Suggestions]: “Please explicitly discuss the limitations of MP-Nav, particularly regarding scenarios where poisoning may not be effective.”\
> **Response** 1: There are potentially two limitations.
> First, MP-Nav's success depends on the availability and quality of open-sourced models (such as CLIP, Deepseek, etc). When embeddings are noisy or misaligned due to insufficient training data, MP-Nav’s guidance may become suboptimal. \
> Second, the limitation (not specific to MP-Nav but to poisoning attacks in general) arises in datasets where the benign samples outnumber poisoned instances that can dilute the poisoning effects. Kindly note that our MP-Nav requires less number of poisoned instances than existing attack methods.
>
> 2 [Claims And Evidence]: “The claim that MP-Nav preserves model utility requires further clarification, as the poisoned data could have long-term effects on model behavior.”\
> **Response** 2: In our experiments, we primarily assessed immediate model performance post-poisoning. Indeed, it is possible that the cumulative influence of poisoning could manifest more over extended fine-tuning or continuous learning. In future work, we will investigate the poisoning effect in continuous learning with a presumed research question on how to enhance the cumulative poisoning effect under catastrophic forgetting in the setting of continuous learning.
>
> 3 [Other Strengths And Weaknesses, Questions For Authors]: “Limited exploration of countermeasures.”  “ Could existing outlier detection or data sanitization methods be used to identify and filter poisoned samples before model training”\
> **Response** 3: We have implemented a simple defense method that is also used as the pre-training defense in [1].
> Since the AtoB is a dirty-label poisoning attack, the input sanitization is an effective countermeasure.
> We used an open-sourced model to calculate cosine similarities of embeddings of both images and their corresponding texts. A higher cosine distance means images and texts are less relevant. We set threshold as 0.8 to avoid benign instances to be filtered out. We report the results below.
>
> |Poison setting | Defense | Poison data Number|
> | -------- | ------- |------- |
> | boat2dog | No input sanitization | 284 (out of 119387) |
> | boat2dog | pre-training defense [1] | 39 |
> | boat2kit (MP-Nav) |  No input sanitization | 284 (out of 119387) |
> | boat2kit (MP-Nav)  |  pre-training defense [1] | 140 |
>
> As we can see above, input sanitization is quite effective in A2B attacks [1].
> Kindly note that MP-Nav makes A2B have stronger resistance against input sanitization.
>
> [1] Data Poisoning Attacks Against Multimodal Encoders, in ICML 2023.
>
> In terms of ShadowCast attack (clean-label attack), our paper has revealed that clean and benign examples can largely mitigate the poisoning effect. Nevertheless,  the MP-Nav can still enhance the poisoning effect under the presence of benign examples.
>
>
> 4 [Questions For Authors]: “Have authors observed any situations where MP-Nav fails to improve poisoning efficacy, or where it inadvertently lowers the effectiveness of an attack compared to baseline methods?”\
> **Response** 4: Indeed, we have observed scenarios where MP-Nav does not outperform baseline methods in some cases. This is due to inherent noise occurring in the training data (such as irrelevant features in the images, noisy captions, data shortage). As visualized in the top two panels of Figure 5 in the appendix, the poisoning effect is noisy w.r.t. similarity scores. Despite the noise, the bottom two panels of Figure 5 unveil the positive correlation between similarity scores and the poisoning effect.
>
>
> 5 [Questions For Authors]: From my understanding, MP-Nav currently assumes access to embeddings for similarity computation. How well does it perform if the attacker does not have access to the model’s internal representations?\
> **Response** 5: Kindly allow us to clarify the assumptions in the paper. The attacker does not have access to the learner’s model (and therefore embeddings) but can get access to an open-sourced surrogate model that computes and compares data’s similarities. Thus, all the reported results reflect scenarios where the attacker does not have access to the model’s internal representation.

---

> > ### Comment · Reviewer_cLuX · 2025-04-04
> >
> > Thank you for your comprehensive response. The authors have provided effective clarifications to my questions, and I have raised my score to 4.

---

> > > ### Author Response · Authors · 2025-04-07
> > >
> > > Thank you for carefully reading our response.

---

### Official Review · Reviewer_sXf2 · 2025-03-14

**Overall Recommendation:** 3

**Summary:**

This paper addresses the vulnerability of large-scale multimodal learning models to data poisoning attacks, where adversaries subtly inject malicious instances into training data to misalign concepts. It proposes MP-Nav (Multimodal Poison Navigator), a module that strategically selects semantically similar concept pairs and robust instances to enhance the effectiveness of poisoning attacks. Experimental results demonstrate that MP-Nav improves attack success rates while preserving model utility, highlighting the security risks of multimodal models and emphasizing the need for stronger defenses.

**Claims And Evidence:**

This paper has 4 major claims.

C1. Not all concepts are equally vulnerable to disassociation.
- I did not see direct evidence for the claim. However, based on an ablation study, the claim may be true.

C2. Not all instances contribute equally.
- I did not see direct evidence for the claim. However, based on the ablation study, the claim may be true.

C3. MP-Nav Enhances Attack Effectiveness: The proposed MP-Nav module systematically selects semantically similar concept pairs and robust instances, significantly improving the success rate of multimodal data poisoning attacks.
- In Section 4.1, the empirical results show that selecting instances and concepts can improve attack effectiveness.

C4. Resilience Against Benign Data Dilution: By selecting robust instances within chosen concepts, MP-Nav ensures that poisoned instances remain effective even when mixed with a large number of benign samples, maintaining attack efficacy while preserving model utility.
- Table 3 shows the utility of the method, which has a marginal difference as the baselines. I doubt if the testing is hard enough, as all methods share similar utility. It is not clear if this attack method will balance utility with attack effectiveness. Since the number of poisoned examples is not specified, it is hard to associate the attack result in Fig 2 with Table 3.

**Essential References Not Discussed:**

No.

**Experimental Designs Or Analyses:**

It is hard to find where the claims 1 and 2 are validated. I can only infer that the ablation study in Sec 3.1 may imply claim 1 and 2.

**Methods And Evaluation Criteria:**

The paper proposes MP-Nav (Multimodal Poison Navigator), a module that strategically selects semantically similar concept pairs and robust instances to enhance the effectiveness of poisoning attacks.

The method enhanced existing attacks AtoB and ShawdowCast while maintaining utility. The paper used Flickr-PASCAL and COCO datasets for evaluation. The metrics are
1. Model Utility – This checks how well the model retrieves the correct results. It uses Recall at K (R@K), which tells us how often the correct answer appears in the top K results when searching for images or captions.
2. Poisoning Efficacy – This measures how well an attacker can trick the model into linking the wrong concepts. It uses Hit@K, which shows how often the wrong (targeted) image appears in the top K results, and MinRank, which tells how early the wrong image appears in the ranked list (lower is worse).

Both metrics are reasonable from the literatures.

**Other Comments Or Suggestions:**

"image captaining" in Line 12.
Figues 2 should be Figure 2 in Line 331.

**Other Strengths And Weaknesses:**

Strength:
* The proposed method improve the existing attacks by selecting concept and instances. The method is simple and effective.

Weakness
* The effectiveness of the method lies in a small range of the number of poisoned samples 10-25. This makes the result less significant. I doubt whether the method is really necessary. An attacker can simply increase the number of poisoned instances can enhance the attack. I did not see a reason not to scale up the poisoned instances. Especially, when 25 samples could be very effective.

**Questions For Authors:**

* What is the intuition for claim 1 and claim 2? Any theoretical insights for the two claims?

**Relation To Broader Scientific Literature:**

The method enhanced existing methods like AtoB and ShadowCast. This method enforces the embeddings’ misalignment in the learned space to achieve the misassociation while preserving the correct alignments for other concept pairs. By selecting data and concepts, the AtoB and ShadowCast methods are strengthened.

**Theoretical Claims:**

No theoretical claims.

---

> ### Author Rebuttal · Authors · 2025-03-27
>
> Many thanks for the reviewer’s statement that “Experimental results demonstrate that MP-Nav improves attack success rates while preserving model utility”, and the acknowledgment that “the method is simple and effective”. Kindly find our response below.
>
> 1 [Claims and Evidence (First two points)]: “It is hard to find where the claims 1 (C1) and 2 (C2) are validated. What is the intuition for claim 1 and claim 2? Any theoretical insights for the two claims?”\
> **Response 1**:
> In the review, the C1 refers to "Not all concepts are equally vulnerable to disassociation," and the C2 refers to "Not all instances contribute equally."
> Kindly note that C1 and C2 are not our claims, but facts/observations that are backed by prior literature such as [1, 2, 3].
> To choose robust instances to enhance poisons, we are inspired by [4].
> Motivated by the facts C1 and C2, we proposed the method of MP-Nav.
> Thanks for pointing out this; we will revise the paper and further clarify the explanations. \
> [1] Geometry-aware instance-dependent adversarial training, in ICLR 2021\
> [2] Data-efficient backdoor attack, in IJCAI 2022\
> [3] Towards effective clean label backdoor attacks, in Pattern Recognition 2023\
> [4] BadLabel: A robust perspective on evaluating and enhancing label-noise learning, in TPAMI 2024
>
> 2 [Claims and Evidence (Comment C4)]: “Table 3 shows the utility of the method, which has a marginal difference as the baselines. I doubt if the testing is hard enough, as all methods share similar utility. It is not clear if this attack method will balance utility with attack effectiveness. Since the number of poisoned examples is not specified, it is hard to associate the attack result in Fig 2 with Table 3.” \
> **Response 2**: In Table 3, we have chosen the standard benchmark test sets, Flickr-PASCAL has a 1K test set, and COCO has a 3.9K  test set. The test set is large and comprehensive for evaluating overall utility. Attacker aims to maintain the model utility and enhance attack efficacy. In Flickr-PASCAL with a training set (around 30K), we only allow 25 poisoned data, and in COCO training dataset (around 120K), we only allow 284 poisoned data.  Marginal difference or even no difference in Table 3 is exactly what the attacker is aiming for. Table 3 (model utility) associates with both Table 2 and Figure 2 (attack efficacy), which justifies the MP-Nav outperforming the baselines. We will revise the paper and make the explanation clear.
>
> 3 [Weakness comment]: “The effectiveness of the method lies in a small range of the number of poisoned samples 10-25. This makes the result less significant. I doubt whether the method is really necessary. An attacker can simply increase the number of poisoned instances can enhance the attack. I did not see a reason not to scale up the poisoned instances. Especially, when 25 samples could be very effective.”\
> **Response** 3: Conceptually, the number of poisoned instances is related to the attacker budget: a more effective attack method prefers using less budget. For example, in decentralized learning, the attacker aims to control as few as possible nodes to conduct effective poisoning attacks. Moreover, injecting a larger number of poisoned instances increases the risk of detection (that is even the case for other attacks -- such as backdoors --  in existing literature). We will clarify this point further in the revision.
>
> 4 [Clarifications & Typos]: “Fig. 5 and 6 are hard to read” and “image captaining in Line 12” “Figues 2 should be Figure 2 in Line 331.”\
> **Response** 4: Figures 5 and 6 are comprehensive results that back Table 2. Main results that justify the efficacy of MP-Nav.  Thank you, we will re-work the figures to enhance readability, add more descriptions, and correct the mentioned typos.

---

### Decision · Program_Chairs · 2025-05-01

**Decision:**

Accept (poster)

**Comment:**

This paper proposes MP-Nav (Multimodal Poison Navigator), a plug-and-play module designed to evaluate and enhance data poisoning attacks against multimodal models. MP-Nav operates at both the concept and instance levels, identifying semantically similar concept pairs and selecting robust instances to maximize the attack efficacy. Experimental results demonstrate that MP-Nav can significantly improve the efficacy of SOTA poisoning attacks in multimodal tasks, and maintain model utility across diverse datasets.

After the rebuttal process, this paper finally receives 2 weak accept and 2 accept. All the reviewers reach an agreement that this paper can be accepted. I also feel that this paper is good work on enhancing data poisoning attacks against multimodal learning, which can underscore the vulnerabilities of multimodal models and urge the corresponding defense methods.